# Synthesis of Glycerol Carbonate from Ethylene Carbonate Using Zinc Stearate as a Catalyst: Operating Conditions and Kinetic Modeling

**DOI:** 10.3390/molecules28031311

**Published:** 2023-01-30

**Authors:** Mariana Alvarez Serafini, David Gonzalez-Miranda, Gabriela Tonetto, Félix Garcia-Ochoa, Miguel Ladero

**Affiliations:** 1Department of Chemical Engineering, Universidad Nacional del Sur (UNS) and Chemical Engineering Pilot Plant—PLAPIQUI (UNS-CONICET), Bahía Blanca B8000, Argentina; 2FQPIMA Group, Materials and Chemical Engineering Department, Chemical Science School, Complutense University of Madrid, 28040 Madrid, Spain

**Keywords:** glycerol valorization, ethylene carbonate, glycerol carbonate, zinc stearate

## Abstract

With the advent of biodiesel as a substitute/additive for diesel, the production of glycerol has experienced an increase, as it is an unavoidable byproduct of the biodiesel process; therefore, novel products and processes based on this triol are being very actively researched. Glycerol carbonate emerges as an advanced humectant from glycerol and a monomer for diverse polycarbonates. Its production in high yields and amounts can be achieved through the solventless transcarbonation of glycerol with other organic carbonates driven by alkaline catalysts, standing out amongst the cyclic carbonates due to its reactivity. Here, we have studied the main operational variables that affect the transcarbonation reaction of glycerol and ethylene carbonate catalyzed by zinc stearate: catalyst concentration, reagent molar ratio, and temperature. Subsequently, an appropriate kinetic model was fitted to all data obtained at 80 °C and several catalyst concentrations as well as reagent molar ratios. Finally, the selected kinetic model was extended and validated by fitting it to data obtained at several temperatures, finding that the activation energy of this reaction with this catalyst is around 69.2 kJ·mol^−1^. The kinetic model suggests that the reaction is bimolecular and elemental and that the process is interfacial in essence, with the catalyst dispersed in a narrow space between polar (glycerol) and nonpolar (ethylene carbonate) phases.

## 1. Introduction

Global biodiesel production has increased over the years, generating large quantities of the byproduct glycerol (Gly), such that glycerol is essentially produced by transesterification nowadays and is not only an ingredient in the food, cosmetic, and pharmaceutical industries, but also a solvent and platform chemical in the chemical industry. The commercial price of glycerol has fallen dramatically, mainly that of technical grades that do not need extensive purification. This fact has a direct impact on the biodiesel industry, to the point that low-quality glycerol is even treated as a waste [1]. Many processes have been proposed for the valorization of glycerol as a byproduct of biodiesel [2,3]. This valorization can be carried out through different routes: thermal, chemical, enzymatic, and by employing diverse microorganisms [4,5] The targeted products can be either ingredients of cosmetics, such as dihydroxyacetone (DHA), or food applications (emollients based on glycerol, or ingredients for functional foods such as modified triglycerides), or they can be platform chemicals used as monomers or low molecular additives in the polymer industry (such as glycidol or glycerol carbonate) [6,7].

Within the chemical routes and processes, the synthesis of glycerol carbonate (GC) stands out. This carbonate, as with the others compounds of this family, allows for the chemical activation of CO_2_ and is a compound of low toxicity, is biodegradable, and has multiple applications. In particular, CC is used as a humectant and a green solvent with a high boiling point. Moreover, it has been tested as a novel component of gas separation membranes, in addition to as a surfactant, as a component of coatings, paints, and detergents, and as a source of monomers for the production of polycarbonates and polyurethanes, and also of glycidol as a less hazardous monomer, replacing epichlorohydrin [2,3,4].

Several routes of glycerol carbonate synthesis have been proposed, such as the direct addition of CO and CO_2_ to glycerol, the addition of CO_2_ to glycidol, or the glycerolysis of urea [4]; however, the most interesting, and productive, synthetic route is the reaction of glycerol with organic carbonates. When glycerol reacts with ethylene carbonate (EC) under alkaline conditions at low to mild temperatures (see Figure 1), glycerol carbonate and ethylene glycol are obtained. Monoethylene glycol, the byproduct, is notably useful in a number of industries: energy, chemical, textile, automotive and transportation, etc. [6].

To carry out this transcarbonation reaction, several basic catalysts have been tested, among which the following can be mentioned: KOCH_3_ [5], K_2_CO_3_ [8], CaO [9], and Al–Mg as well as Al–Ca hydrotalcites [10]. Potassium methoxide is a strong basic catalyst that shows very high activity in biphasic media at 40–50 °C at 50–150 ppm catalyst in solventless conditions, as it concentrates in the glycerol phase [5]. Potassium carbonate is also a homogeneous catalyst with lower basicity; reactions proceed at mild temperatures in solventless conditions and higher catalyst concentrations (till 1.25% *w*/*w* of glycerol) [8]. Classical heterogeneous catalysts are calcium oxide and Al-Mg hydrotalcites. CaO performed well under the following reaction conditions: GL: 9.2 g; DMC/Gly molar ratio: 3; reaction temperature: 80 °C; and Cat/GL molar ratio: 0.15 (in CaO weight). At a reaction time of 5 h, glycerol conversion was more than 95% while the carbonate yield was 90.57%; however, it shows a sharp deactivation that is surely due to water action. As for the hydrotalcites, using a two to one molar ratio of EC to glycerol, 7% (*w*/*w*) of catalyst with respect to the total weight of reactants, and a temperature of 50 °C, the yield to glycerol carbonate was 82% with only minor amounts of glycidol at 2 h [9]. Deactivation is evident after only three cycles. Considering metallic soaps, zinc carboxylate was studied as an active catalyst in the triglyceride transesterification reaction to produce biodiesel [11]. With this catalyst, triglyceride conversion reached values higher than 80% at 60 min, 5% catalyst concentration, and a 10–30 methanol/oil molar ratio; FAMEs yield was near 60% at those conditions, but only at the highest methanol/oil molar ratio. This catalyst was also used as a catalyst for the esterification of fatty acids (a commercial mix of oleic acids) with ethanol, as was zinc laurate, reaching acid conversions up to 96% in 90 min [12]. In that case, the catalyst could be recovered after the reaction as a powder via simple filtration and used again with no activity loss due to its layered structure. Zinc stearate is also useful when employed in free-fatty-acid-rich olive pomace oil, reaching very high conversions if glycerol and water are progressively removed [13,14]. Alkali metal carboxylates have also been shown to be highly active as catalysts for the ring-opening polymerization of cyclic esters [15,16]. To our best knowledge, however, there are no reports on the activity of metal fatty acid salts acting as catalysts to produce glycerol carbonate or, in fact, in any transcarbonation reaction.

Kinetic modeling is key to the simulation of processes under similar, but not identical, conditions to those studied experimentally, using various forms of operation, as well as being the basis for the design of chemical reactors and facilitating their control in operation. The synthesis of glycerol carbonate has been shown to be an elemental and bimolecular reaction between glycerol and other carbonates; the deactivation of catalysts has been observed, which can be seen in the kinetic models obtained by fitting their integrated ordinary differential equations to the experimental data over a wide range of experimental conditions [5,8].

In this work, we have explored the activity of a metal soap, zinc stearate, as the catalyst of the reaction between glycerol and ethylene carbonate in the absence of solvents, producing the targeted glycerol carbonate along with a very interesting byproduct: monoethylene glycol. The effect of different operating conditions, such as the molar ratio of reactants, catalyst loading, and operating temperature, on the glycerol conversion will be evaluated. Once these variables have been optimized, a kinetic model will be proposed to describe the temporal evolution of the key compounds’ compositions in the reacting system.

## 2. Results and Discussion

As a preliminary experiment, a blank test was performed. The conversion of glycerol after 6 h at 80 °C in the absence of a catalyst was measured, and less than 5% conversion was observed. Therefore, there is an evident need for a catalyst for the reaction to proceed at an appreciable rate, and the thermal contribution can be neglected.

### 2.1. Mass Transfer Resistance

The transesterification reaction occurs in a liquid–liquid biphasic system where there is a liquid phase rich in glycerol and another one rich in ethylene carbonate. The catalyst, due to its amphiphilic nature, should tend to stay in the interphase, as happens with a soybean oil–methanol system for biodiesel production [12]. As the glycerol is the less abundant compound, the phase rich in this polyol is dispersed in the carbonate phase. To check the possible external mass transfer resistance, experiments with 3% catalyst relative to the weight of glycerol, a molar ratio of EC:Gly 2:1, and a temperature of 80 °C were carried out at different stirrer speeds (from 600 to 800 rpm) to determine the conditions in which the external mass transfer resistance does not limit the overall process rate. These experiments show that the observed initial reaction rate, 0.020 ± 0.0015 mol/(L·min), was independent of the agitation rate above 700 rpm (data not shown). Hence, all further experiments were carried out at a stirrer speed of 800 rpm to ensure that there was no significant external mass transfer resistance.

Although the concentrations of the reaction components were obtained by ion-exclusion chromatography, we can see in Figure 2 an ^1^H-NMR spectrum of a sample at 200 min when using a catalyst concentration of 3% *w*/*w* glycerol, a CE/Gly ratio of 2, and a temperature of 80 °C. We can appreciate the presence of signals at diverse δ values corresponding to the four main components of the reaction system, while no other signals are present, indicating (as perceived in the chromatograms) an almost exquisite selectivity to the target products.

### 2.2. Effect of Catalyst Loading in Glycerol Conversion

To study this variable, three experiments were carried out at 80 °C, using a molar excess of ethylene carbonate to glycerol (reagent molar ratio equal to two) and setting the catalyst concentration at values 1, 2, and 3% *w*/*w* of glycerol.

The conversion of glycerol obtained in these three runs is given in Figure 3A. As can be seen, a maximum conversion of around 77% was obtained at 6 h of reaction time with both 3 and 5% catalyst loading, greater than that obtained with lower catalyst loading of 1%. Figure 3B shows the variation in the initial reaction rate with the catalyst concentration. There is a significant increase in this variable with the increase in catalyst load between 1 and 3%, and a lower increase between 3 and 5%. The hyperbolic trend towards an asymptotic value of the reaction rate would indicate that the reaction takes place at the liquid–liquid interface, since an emulsion is observed to form when the catalyst is added. The emulsion formation is attributed to the surfactant nature of the catalyst (a metallic soap). Therefore, it is hypothesized that zinc stearate tends to be in this liquid–liquid interface, which is only able to accommodate a certain amount of the catalyst. At higher values, the catalyst molecules would not be able to interact with either glycerol or ethylene carbonate, or such an interaction would be much limited by liquid-liquid mass transfer in cases where the non-available reactant is able to go beyond the interface where a chemical reaction is happening at a fast rate due to the high local concentrations of reactants and catalysts.

Considering these results, catalyst loading of 3% has been considered the most suitable in terms of catalytic activity and was selected for use in further experiments considering the other operational variables.

### 2.3. Effect of the Initial Molar Ratio of Reactants

A second set of runs consisted of four experiments that were performed that changed the CE/Gly molar ratio, employing values of 1.5, 2, 2.5, and 3. These experiments were also carried out at 80 °C and, as commented before, set the catalyst load at 3% *w*/*w* of glycerol.

Figure 4A shows the conversion of glycerol with time obtained in these runs. The maximum conversion of around 77% was obtained at 6 h of reaction time with the highest molar ratio studied, as expected when the reaction is shifted towards the products by a high excess of ethylene carbonate; however, it can be observed that, at short times, the fastest reaction occurs when the CE/Gly molar ratio is equal to 2. This suggests that a big excess of carbonate means excessive dilution of glycerol, negatively affecting the global reaction rate. This observation can be better appreciated in Figure 4B, where the relationship between the initial reaction rate and the molar ratio of reactants is shown. As can be seen, there is an optimum value with the molar ratio of CE/Gly = 2. In view of these results, runs to study the effect of temperature were performed with an ethylene carbonate to glycerol molar ratio of two.

### 2.4. Effect of the Temperature

The influence of temperature on the conversion of glycerol was studied under the conditions mentioned above: an initial CE/Gly molar ratio of two and catalyst loading of 3%.

It is important to mention that at low temperature and in the absence of the Zn stearate, between 60 and 70 °C, the reactants are not miscible, meaning that two different liquid phases are present at the beginning of the process in the reactor. As the reaction proceeds and products are formed, a monophasic system is reached [4,8]. On the other hand, at 80 and 90 °C glycerol and ethylene carbonate are apparently miscible, but the addition of the catalyst again creates an emulsion, a liquid–liquid system.

The results of glycerol conversion at different temperatures as a function of reaction time are shown in Figure 5A. The increase in the initial reaction rate (r_0_) with the increase in temperature can be seen in Figure 5B. In that last figure, it is clearly observed that there is an exponential increase in the initial reaction rate with temperature, following the Arrhenius trend. Figure 6 shows the dependence of ln (r_0_) with the inverse of the absolute temperature 1/T; a linear tendency can be appreciated. The slope of the line is used to determine an approximate activation energy of the studied reaction, obtaining a value of 37.35 kJ/mol. This value of activation energy is within the range of expected values, taking into account what is reported in the literature. Devi et al. obtained an activation energy of 39.2 kJ/mol for the reaction between glycerol and dimethyl carbonate using a heterogeneous catalyst, Ti-SBA-15 [17]. Values between 72 and 92 kJ/mol were found for the reaction of glycerol and ethylene carbonate with a homogeneous catalyst, K_2_CO_3_ [8].

### 2.5. Kinetic Modeling

Transesterification and transcarbonation reactions are usually elemental bimolecular reactions when using catalysts in both a homogeneous liquid phase and in a multiphasic liquid system; therefore, the partial orders with respect to any of the chemical compounds of the reaction system is one [4,8]. In addition, in homogeneous reaction systems the effect of the catalyst concentration in the reaction rate is linear as the contact between reagents and the catalyst is free, without any limitation due to mass transfer or to any spatial hindrance. This is not the case for heterogeneous systems, where mass transfer limitations in pores are expected when heterogeneous catalysts are employed, and the reagent–catalyst contact is restricted to one of the fluid phases or to the liquid–liquid interphase [18,19]. Finally, the effect of reactants that conform a biphasic system due to their relative insolubility creates an emulsion where the ratio of phase volumes determines the interfacial area, thus affecting the reaction rate; the interfacial area usually increases as the discontinuous phase volume increases, usually in a hyperbolic or sigmoidal manner [14,15,20].

Accordingly, three kinetic models have been proposed, accepting that the effect of any of the chemical compounds involved in the reaction is linear (partial order = 1), with the exception of the catalyst concentration, whose effect on the reaction rate is seen to be hyperbolic or potential. Model 1 assumes that this effect is potential and assigns a random order n to the catalyst concentration (Equation (1)). Model 2 accepts that the catalyst has a hyperbolic effect on the reaction rate, such that it is disposed at the interface between glycerol and ethylene carbonate until this interface is saturated with the catalyst, emulating a behavior similar to that of a reagent adsorbing on a surface following the Langmuir adsorption model (Equation (2)). The third model considers how the phases are distributed in the two-phase system at zero time: glycerol is the dispersed or discontinuous phase in all cases, and even more so the higher the carbonate excess. Model 3 assumes that the reaction rate changes in a sigmoidal way with the molar ratio between ethylene carbonate and glycerol, maintaining the hyperbolic effect of the catalyst concentration on the reaction rate (Equation (3)). The equations of the different models cited are as follows:(1)$$\frac{{\mathrm{dX}}_{\mathrm{Gly}}}{\mathrm{dt}}=k\cdot C_{\mathrm{cat}}^{n}\;\cdot {\;C}_{\mathrm{Gly},\;0}\cdot \left(1-X_{\mathrm{Gly}}\right)\cdot \left(M-X_{\mathrm{Gly}}\right)$$
(2)$$\frac{{\mathrm{dX}}_{\mathrm{Gly}}}{\mathrm{dt}}=k\cdot \frac{C_{\mathrm{cat}}}{K_{\mathrm{cat}}+C_{\mathrm{cat}}}\;\cdot {\;C}_{\mathrm{Gly},\;0}\cdot \left(1-X_{\mathrm{Gly}}\right)\cdot \left(M-X_{\mathrm{Gly}}\right)$$
(3)$$\frac{{\mathrm{dX}}_{\mathrm{Gly}}}{\mathrm{dt}}=k\cdot \frac{C_{\mathrm{cat}}}{K_{\mathrm{cat}}+C_{\mathrm{cat}}}\;\cdot \frac{M^{n}}{K_{M}^{n}+M^{n}}\;\cdot C_{\mathrm{Gly},\;0}\cdot \left(1-X_{\mathrm{Gly}}\right)\cdot \left(M-X_{\mathrm{Gly}}\right)$$

These three kinetic models have been tested for the transesterification reaction of glycerol with ethylene carbonate, using all of the data of the experiments carried out at the same temperature (80 °C). In all cases, the reaction is accepted to proceed until the total conversion of glycerol (or very near it). Good liquid–liquid mixing is ensured as all runs are performed at a high stirring rate (700 rpm), and the temperature is tightly controlled at the aforementioned value by a PID system.

Table 1 compiles the values of the kinetic parameters together with their standard errors at 95% confidence, while Table 2 shows the values of the goodness-of-fit statistical parameters for the three models. It can be appreciated that all of the kinetic parameters are positive and that their standard errors are low in comparison to the values of the parameters, so they all pass the *t*-test. As for the goodness-of-fit parameters, it can be observed that model 3 leads to the lowest values of the residual sum of squares (RSS) and the standard error of estimate (S_e_) in addition to the highest values of Fisher’s value and the variation explained (VE). Therefore, model 3 is clearly the best one, and this can also be appreciated in the good fitting of the model to all data at 80 °C (see Figure 7).

### 2.6. Fitting of the Chosen Kinetic Model at Several Temperature Data

Once the kinetic model was selected, the four runs performed at temperatures from 60 to 90 °C were used to validate it and define the kinetic constant as an exponential function of such a relevant variable. To this end, the K_cat_ value was set at 10.26, K_M_ was 1.47, and n (for model 3) was set at 9.67, defining k as an Arrhenius function according to the following equation:(4)$$k=exp\left(k_{0}-\frac{E_{a}}{R}\frac{1}{T}\right)$$

Results of the fit can be observed in Figure 8. Table 3 compiles the relevant parameters of the kinetic constant together with goodness-of-fit statistical parameters. While we can appreciate that the fit of the model to data at all temperatures is excellent, the optimal value of the activation energy estimated is 69.1 kJ/mol, well within the interval of values found in the literature for this reaction. Therefore, the kinetic model representing the behavior of the reacting system under the tested conditions is as follows:
(5)$$\frac{dX_{Gly}}{dt}=exp\left(12.05-6623\frac{1}{T}\right)\cdot \frac{C_{cat}}{10.26+C_{cat}}\;\cdot \frac{M^{9.67}}{1.47^{9.67}+M^{9.67}}\cdot \;C_{Gly,\;0}\times \left(1-X_{Gly}\right)\cdot \left(M-X_{Gly}\right)$$

Zinc stearate is notably less active than K_2_CO_3_ [21] and KOCH_3_ [4]. Potassium catalysts are homogeneous polar catalysts that able to be easily solved in the glycerol phase; at a glycerol/ethylene carbonate molar ratio (M) of two, K_2_CO_3_ showed a TOF value of 0.95 s^−1^ at 40 °C, while potassium methoxide TOF was 0.27 s^−1^ at 70 °C. At that same high temperature and molar ratio, Zn stearate showed a TOF of 1.5 × 10^−3^ s^−1^; however, it should be noted that potassium catalysts suffered from deactivation, while Zn carboxylate showed no sign of instability in the experimental interval, as indicated by the selected and validated kinetic model. Although there is no literature regarding the reaction with ethylene carbonate with TOF data for heterogeneous catalysts, Alvarez et al. estimated TOF values in 6.7 × 10^−5^–3.3 × 10^−2^ for diverse hydrotalcites using diethylcarbonate and glycerol [22]. As a conclusion, evident pore mass transfer limitations and/or interfacial effects can be appreciated with heterogeneous and interfacial catalysts.

## 3. Materials and Methods

### 3.1. Materials

Glycerol (99+%, Fisher Chemical, Waltham, MA, USA) and ethylene carbonate (Scharlab EssentQ^®^, >99%, purchased to Scharlab S.L., Barcelona, Spain) were used as reactants. Zinc stearate (extra pure, Pharmpur^®^, Ph Eur, BP, USP, provided by Scharlab S.L., Barcelona, Spain) was used as catalyst.

### 3.2. Experimental Procedure

The experiments were carried out in 50 mL glass reactors operating in a batch mode. The round-bottomed flasks were placed in an aluminum gasket inserted in a heating plate with magnetic agitation and temperature PID control, as shown in Figure 9. The reagents were mixed until the working temperature was reached by using a temperature controller. The catalyst was then added, and the initial time sample was taken and diluted to 1:15 with an aqueous solution of citric acid at 8 g/L, used as the internal standard. The system evolution with time was studied for 6 h.

The operating conditions studied were catalyst loading (in the range of 1 to 3% in mass with respect to glycerol) and the initial molar ratio of ethylene carbonate to glycerol (between 1 and 3); once the optimal values of these variables were determined, the effect of the temperature was studied (from 60 to 90 °C).

The reaction samples at different times were analyzed using HPLC-ion exclusion employing a BP 800-H column (Benson, Reno, NV, USA) in addition to the use, as an eluent, of Milli-Q water acidified with H_2_SO_4_ 0.005 M to obtain a pH of 2.2. The flow of the eluent was controlled at 0.5 mL/min and the temperature of the column was set at 60 °C. For the detection of alcohols and carbonates, we used a refractive index detector set at 45 °C.

Once the peak areas were estimated and the lineal internal standard curves for glycerol at diverse reagent molar ratios were applied, the progress of the reaction was calculated as glycerol conversion, according to the following equation:(6)$$X=\frac{C_{0,Gly}-C_{Gly}}{C_{0,Gly}}$$
where C_0,Gly_ and C_Gly_ are the glycerol concentrations at the initial time and at a certain time, respectively.

Once raw glycerol conversions were estimated for all of the runs, Origin 2021 software was applied to smooth the data through interpolation with hyperbolic functions. Smoothing allows for random experimental error elimination, providing corrected values of glycerol conversion that were the data used for kinetic modeling.

### 3.3. Kinetic Modeling Discrimination and Validation

The kinetic model is explained in the last part of the previous section after observing the influence of the operational variables on the initial reaction rate. We have fitted all of the proposed kinetic models to the experimental data obtained at 80 °C, at several EC:Gly molar ratios from 1.5 to 3, and at catalyst concentrations of 1, 3, and 5% *w*/*w* of glycerol. Fitting was performed by coupling the numerical integration of the ODE of each kinetic model by a variable step Euler method coupled to the flexible gradient method for non-linear regression, called NL2SOLV. The algorithms containing these numerical methods are included in Aspen Custom Modeler v11, a program of the engineering simulation suite Aspen v11. Afterwards, once the most accurate kinetic model was chosen its kinetic constant was expressed by using the Arrhenius equation (Equation (4)), and a second round of model fitting (and validation) was performed by fixing all of the constants but the kinetic constant that is an exponential function of temperature. These multivariable fittings led to a kinetic model that was adequate for the vast experimental range studied.

The fittings of the kinetic models to data at 80 °C and of the selected kinetic models to data from runs at temperatures from 60 to 90 °C were analyzed in terms of goodness-of-fit by using diverse parameters related to the least squares method. This is a statistical consideration, but, evidently, physicochemical criteria were also applied: the positive sign of all thermodynamic as well as kinetic constants and the adequate value of the activation energy (E_a_) in terms of kJ/mol (it should be between 20 and 300 kJ/mol). As commented, all goodness-of-fit statistical criteria applied are based on minimizing the sum of squared residuals (RSE) obtained with the squared values of the difference between the experimental values of glycerol conversions (X_e_) and the values of this variable estimated with the relevant kinetic model (X_c_). The RSE is estimated with Equation (7). The standard error of estimate (SEE) is calculated with Equation (8), while Fisher’s value (F) is calculated with Equation (9). At low values of the RSS and RSE the F-value is large, but it is enough that it is higher than the threshold value for N data and the K parameters of each kinetic models to render the kinetic model significant from a statistical perspective. Such a threshold value depends on the confidence and, usually, is taken at 95% confidence. For the kinetic model discrimination at 80 °C, there are 114 pieces of data and K fluctuates between K = 2 (models 1 and 2) and K = 4 (model = 3), so the F-value thresholds are 19.5 and 5.66, respectively. The fitting of model 3 to data at 4 temperatures was carried out with 76 pieces of data (N) and 2 fluctuating parameters (K), so the critical F-value is 19.5. Therefore, an F-value exceeding these critical values means an adequate kinetic model, while the highest value of F indicates that the model is the best one using a goodness-of-fit parameter that avoids overparameterization.
(7)$$\mathrm{RSE}=\overset{N}{\underset{i=1}{\sum }}{\left(X_{e}-X_{c}\right)}_{i}^{2}$$
(8)$$e=\frac{1}{N}\sqrt{\overset{N}{\underset{i=1}{\sum }}{\left(X_{e}-X_{c}\right)}_{i}^{2}}$$
(9)$$F-\mathrm{value}=\frac{\sum_{n=1}^{N}\frac{{\left(X_{c}\right)}^{2}}{K}}{\sum_{n=1}^{N}\frac{{\left(X_{e}-X_{c}\right)}^{2}}{N-K}}$$

The percentage of variation explained (VE) indicates how well the model predicts the evolution of the dependent(s) variable(s) with the independent variable; in this case, with time. It is expressed by Equation (10) [6]:(10)$$\mathrm{VE}\;\left(\%\right)=100\cdot \left[1-\frac{\sum_{n=1}^{N}{\mathrm{SSQ}}_{i}}{\sum_{n=1}^{N}{\mathrm{SSQ}}_{\mathrm{mean}\;i}}\right]$$

Further mathematical description of this parameter related to the model heteroscedasticity can be found elsewhere [8].

## 4. Conclusions

In the transcarbonatation reaction between glycerol and ethylene carbonate driven by zinc stearate, we can observe a typical hyperbolic trend of the initial reaction rate at increasing amounts of the catalyst, indicating the interfacial nature of the chemical transformation. An excess of the carbonate reagent involves an increase in the final yield, but, at reagent molar ratios higher than two, this excess leads to a decrease in the initial reaction rate, indicating excessive dilution of the glycerol. The trend in the initial reaction rate with temperature is exponential, as expected, while the two-stage kinetic modeling suggests that the reaction is bimolecular as well as elemental and that the process is interfacial in essence, with the catalyst dispersed in a narrow space between polar (glycerol) and nonpolar (ethylene carbonate) phases. Furthermore, a molar ratio of two or higher seems to promote a good dispersion of glycerol in ethylene carbonate, increasing final yields.

## Figures and Tables

**Figure 1 molecules-28-01311-f001:**
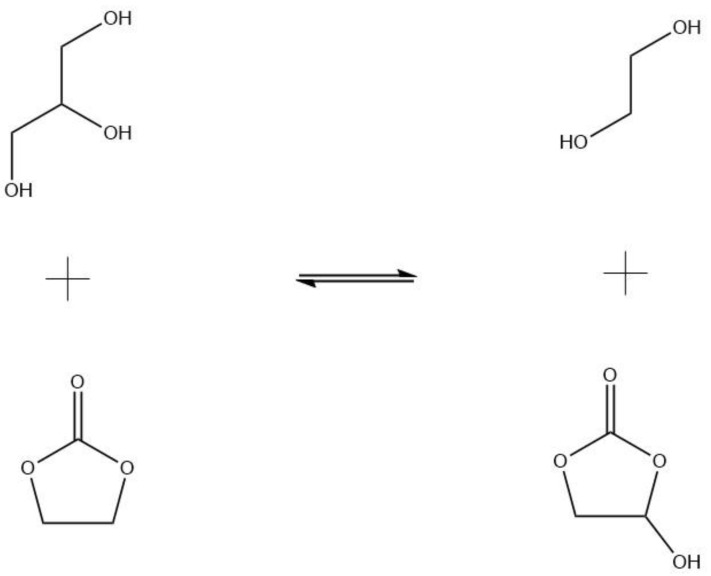
Glycerol transesterification or transcarbonation with ethylene carbonate to render glycerol carbonate and monoethylene glycol (MEG).

**Figure 2 molecules-28-01311-f002:**
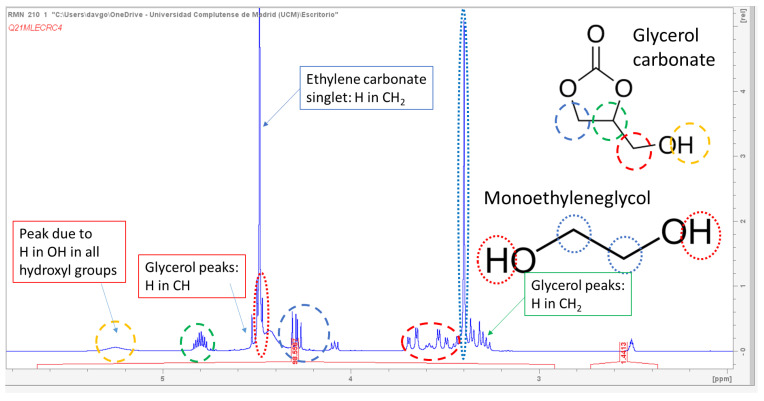
^1^H-RMN spectrum of a representative sample at half conversion showing the main signals due to diverse hydrogen atoms of the main components (alcohols and carbonates) in the reacting system.

**Figure 3 molecules-28-01311-f003:**
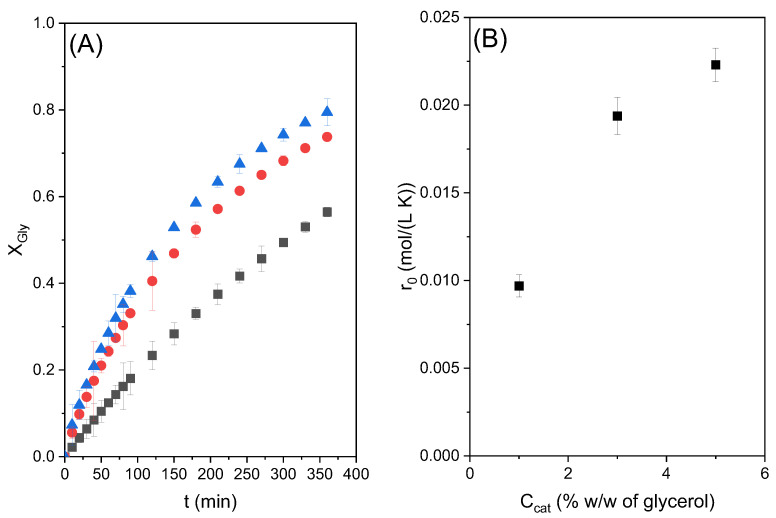
(**A**) Effect of catalyst loading at 80 °C and a CE/Gly ratio of 2 on the conversion of Gly as a function of reaction time with different catalyst loadings, where ■ represents 1% catalyst loading, ● 3% catalyst loading, and ▲ 5% catalyst loading. (**B**) Initial observed reaction rates for these experiments.

**Figure 4 molecules-28-01311-f004:**
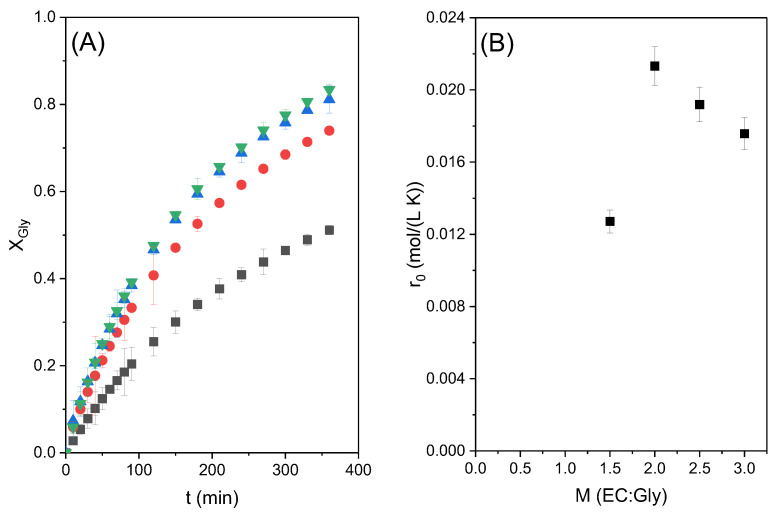
(**A**) Effect of the initial molar ratio of reactants at 80 °C with 3% catalyst loading. In this subfigure the glycerol conversion is presented as a function of reaction time with the CE/Gly ratio (M), where ■ M = 1.5; ● M = 2; ▲ M = 2.5; and ▼ M = 3. (**B**) Effect on the initial reaction rate due to the initial CE/Gly molar ratio.

**Figure 5 molecules-28-01311-f005:**
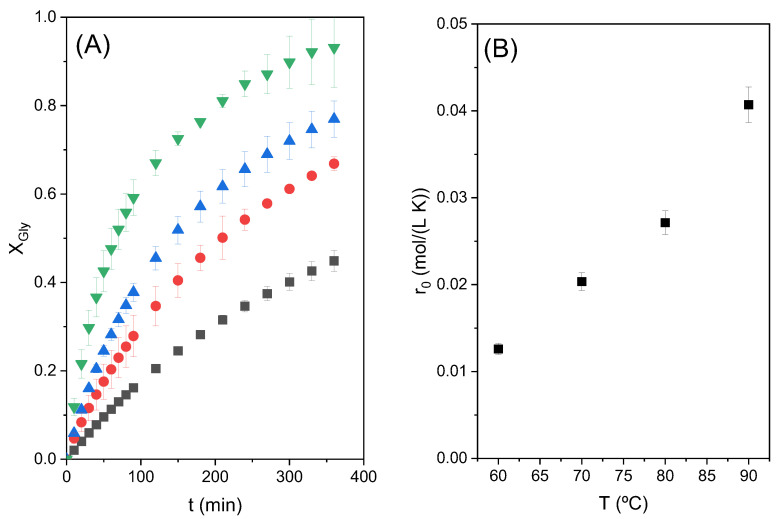
(**A**) Effect of the temperature in the transesterification reaction of Gly with CE with 3% catalyst loading and a CE/Gly molar ratio of 2. In this subfigure ■ represents 60 °C; ● 70°C; ▲ 80°C; and ▼ 90 °C. (**B**) Variation of the initial reaction rate with the different studied temperatures.

**Figure 6 molecules-28-01311-f006:**
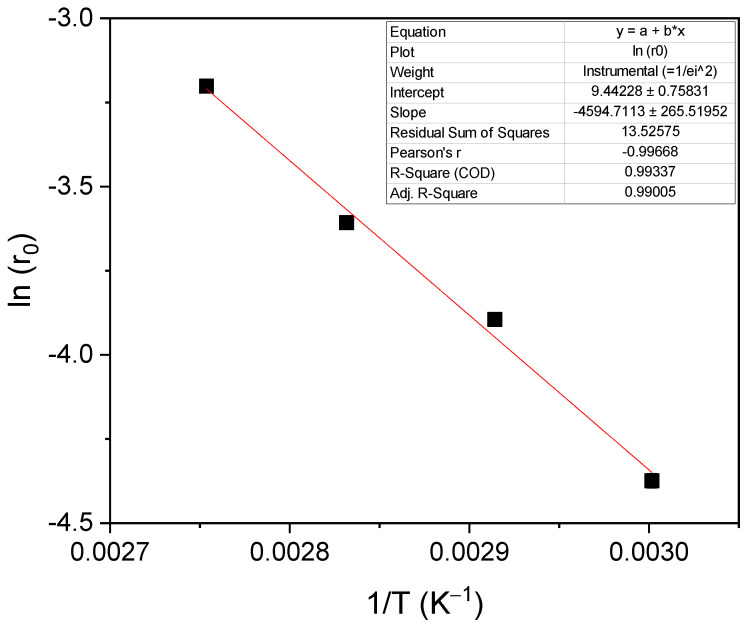
Dependence of the initial reaction rate with temperature (Arrhenius plot).

**Figure 7 molecules-28-01311-f007:**
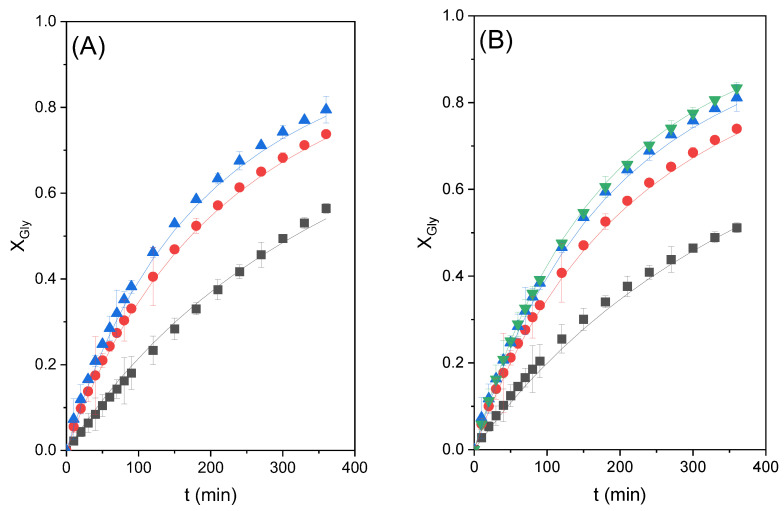
Glycerol conversion (X_Gly_) in the transesterification reaction of glycerol with ethylene carbonate catalyzed by zinc stearate under different conditions. Dots represent experimental data, while lines represent the trend in the model for each piece of experimental data. (**A**) Effect of catalyst loading at 80 °C with a CE/Gly ratio of 2. Gly conversion as a function of reaction time with different catalyst loadings, where ■ represents 1% catalyst loading, ● 3% catalyst loading, and ▲ 5% catalyst loading. (**B**) Effect of the initial molar ratio of reactants at 80 °C with 3% catalyst loading. Gly conversion as a function of reaction time with the CE/Gly ratio, where ■ represents 1.5; ● 2; ▲ 2.5; and ▼ 3.

**Figure 8 molecules-28-01311-f008:**
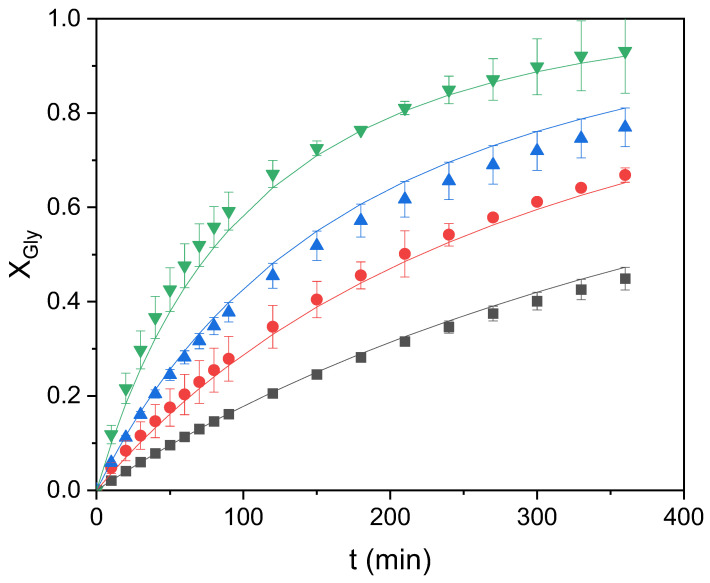
Glycerol conversion (X_Gly_) in the transesterification reaction of glycerol with ethylene carbonate catalyzed by zinc stearate at different temperatures (■: 60 °C; ●: 70 °C; ▲: 80 °C; and ▼: 90 °C).

**Figure 9 molecules-28-01311-f009:**
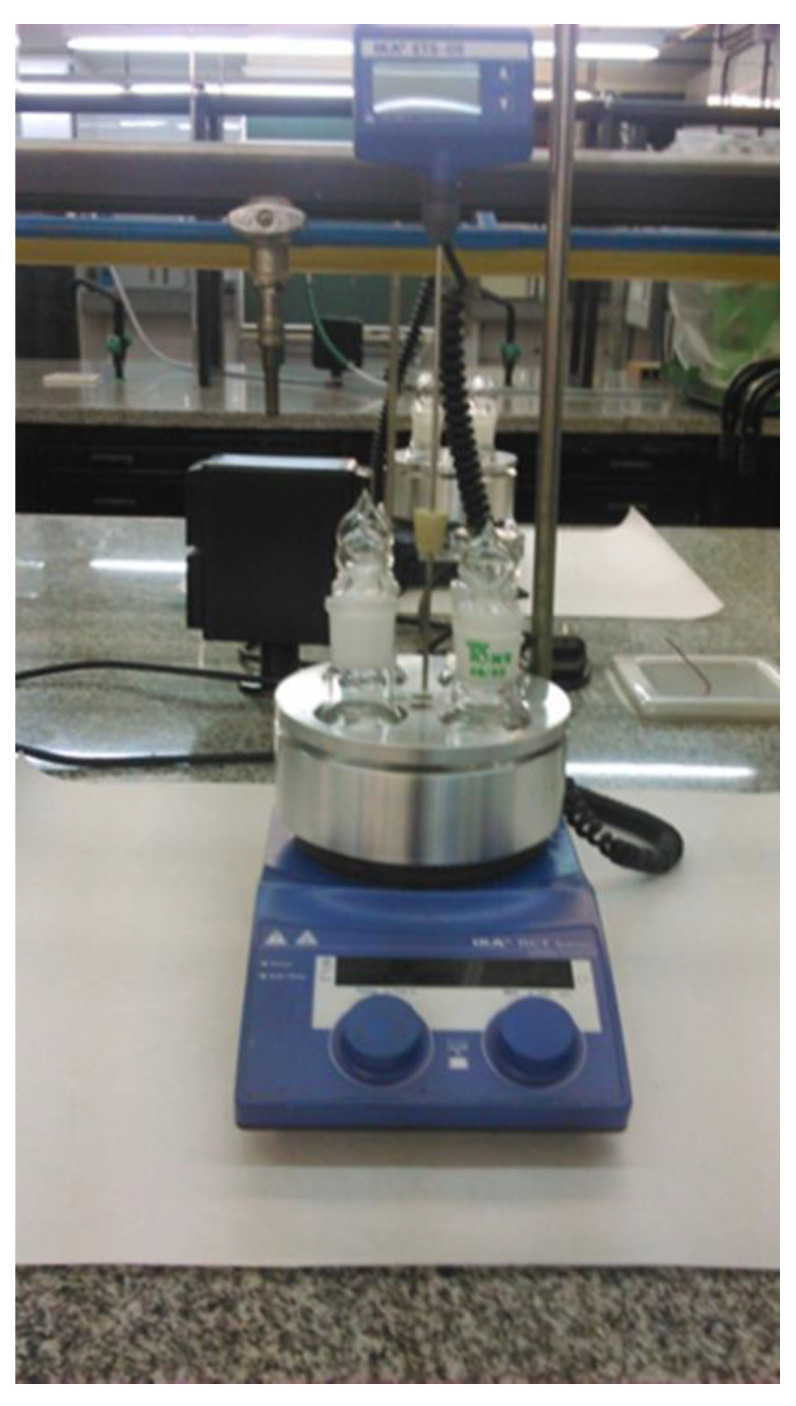
Experimental set-up showing the 50 mL round-bottomed flasks placed in the aluminum gasket inserted in a magnetic stirrer provided with a PID temperature control.

**Table 1 molecules-28-01311-t001:** Values and standard errors of the kinetic constants calculated by fitting the kinetic models to experimental results retrieved at 80 °C and diverse catalyst concentrations as well as reagent molar ratios.

Model	k	n	K_cat_	K_M_
1	1.33 × 10^−4^ ± 1.06 × 10^−5^	0.524 ± 0.031	nd	nd
2	8.38 × 10^−4^ ± 7.16 × 10^−5^	nd	9.60 ± 1.85	nd
3	9.60 × 10^−4^ ± 4.13 × 10^−5^	9.67 ± 1.64	10.26 ± 0.93	1.47 ± 0.012

**Table 2 molecules-28-01311-t002:** Statistical parameters calculated by fitting the kinetic models to experimental results: residual sum of squares (RSS), standard error of estimate (S_e_), percentage of variation explained (VE%), and Fisher’s value (F-value).

Model	RSS	S_e_	VE%	F-Value
1	0.116	0.035	97.91	7921
2	0.183	0.035	97.79	7623
3	0.064	0.024	98.89	9212

**Table 3 molecules-28-01311-t003:** Statistical parameters calculated by fitting the sigmoidal model to different sets of experimental data at varying temperatures. Residual sum of squares (RSS), standard error of estimate (Se), percentage of variation explained (PVE%), and Fisher’s value (F-value).

Model	k_0_	E_a_/R	RSS	S_e_	PVE%	F-Value
3	12.05 ± 0.52	6623 ± 182	0.072	0.068	98.49	8316

## Data Availability

All relevant data are provided in the figures and tables of this manuscript. The authors will provide any interested reader with tables containing data on figures.

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
