# Peer review of "Synthesis of Glycerol Carbonate from Ethylene Carbonate Using Zinc Stearate as a Catalyst: Operating Conditions and Kinetic Modeling"

_molecules, 2023, doi:10.3390/molecules28031311_

Round 1

Reviewer 1 Report

Recommendation: minor revision.

Comments: In this work, the authors explored the activity of zinc stearate as the catalyst of the reaction between glycerol and ethylene carbonate in the absence of solvents, producing glycerol carbonate and mono-ethylene glycol. An appropriate kinetic model was performed on the grounds of results obtained at 80 °C and several catalyst concentrations and reagent molar ratios. At this stage, I recommend that this manuscript can be accepted for publication in Molecules after minor revision.

1. The introduction of this paper needs to make a strong argument about the impact and novelty of the work further. So, the introduction should enrich some related articles in this section.

2. If the authors can provide a photo of the reaction device that would be better.

3. The products glycerol carbonate and mono ethylene glycol are better verified by Nuclear magnetic resonance spectrometer.

4. The zinc stearate catalyst is particular to the reaction between glycerol and ethylene carbonate producing glycerol carbonate and mono-ethylene glycol. If not, the author should compare the performance with other reported catalysts.

5. Some small writing mistakes exist in the manuscript. The authors should carefully check and correct them.

Author Response

First reviewer

We are very grateful to the reviewer for his/her comments on our manuscript. They have helped us notably to present a better second version.

 “Comments: In this work, the authors explored the activity of zinc stearate as the catalyst of the reaction between glycerol and ethylene carbonate in the absence of solvents, producing glycerol carbonate and mono-ethylene glycol. An appropriate kinetic model was performed on the grounds of results obtained at 80 °C and several catalyst concentrations and reagent molar ratios. At this stage, I recommend that this manuscript can be accepted for publication in Molecules after minor revision.”

We are very grateful to the reviewer for his/her fine appreciation of our work.

Comment 1

“The introduction of this paper needs to make a strong argument about the impact and novelty of the work further. So, the introduction should enrich some related articles in this section.”

We have tried to make a more in-depth comment on the importance of the process and of the targeted products, extending descriptions or facts already contained in the papers that we have used in the introduction. Of course, as the subject is of real interest nowadays, there are a high number of references, but we found that the ones we mentioned are a fine representation of the present literature. Now, the reviewer can find the following new texts in the Introduction section:

The second paragraph of the introduction has been enlarged:

“Within the chemical routes and processes, the synthesis of glycerol carbonate (GC) stands out. This carbonate, as others compounds of this family, allows for the chemical activation of CO2 and is a compound of low toxicity, biodegradable, and with multiple applications. In particular, CC is used as a humectant and a green solvent with a high boiling point. Moreover, it has been tested as a novel component of gases separation membranes, also as a surfactant, component of coatings, paints, and deter-gents and as a source of monomers for the production of polycarbonates and polyure-thanes, and also of glycidol as a less hazardous monomer replacing epichlorohydrin [2-4].”

At the end of the third (revised) paragraph:

“…To our best knowledge, however, there are no reports on the activity of metal fatty acid salts acting as catalyst to produce glycerol carbonate or, in fact, in any transcarbonatation reaction.”

Final paragraph of the Introduction section, start of the aim of the work:

“In this work, we have explored the activity of a metal soap: zinc stearate as the catalyst of the reaction between glycerol and ethylene carbonate in the absence of solvents, producing the targeted glycerol carbonate along with a very interesting by-product: monoethylene glycol…”

Comment 2

“If the authors can provide a photo of the reaction device that would be better.”

Thank you for the suggestion. We have added a Figure (Figure 8) in the materials and methods section; it is a very simple device as we only used 50 mL round bottom flasks in an aluminum gasket placed in a heater/magnetic stirrer plate provided with a PID temperature controller and agitation control. Likewise, we have modified the description text as we see that is too concise and have some errors. Now the first paragraph in section 4.2 reads as follows:

“The experiments were carried out in 50 mL glass reactors operating in batch mode. The round-bottom flasks were placed in an aluminum gasket inserted in a heating plate with magnetic agitation and temperature PID control, as shown in the Figure 8…”

Comment 3

“The products glycerol carbonate and mono ethylene glycol are better verified by Nuclear magnetic resonance spectrometer.”

We agree with the reviewer and we have provided a Figure with an explained 1H-RMN spectrum and further explanations in the main text at the beginning of subsection 2.1, at the end of the first paragraph in that section. The new text is:

“…Although the concentrations of the reaction components were obtained by ion-exclusion chromatography, we can see in Figure 2 a 1H-NMR spectrum of a sample at 200 min when using a catalyst concentration of 3% w/w glycerol, a CE/Gly ratio of 2 and a temperature of 80 °C. We can appreciate the presence of signals at diverse d values corresponding to the four main components of the reaction system, while no other signals are present, indicating (as perceived in the chromatograms) an almost exquisite selectivity to the target products.”

In any case, our experience with NMR and FTIR in this reaction system agrees well with our findings in HPLC. Thus, all five main peaks in the chromatograms are clearly due to the reagents, products and internal standard (citric acid) used. In fact, at all experimental time values and diverse reaction conditions, glycerol conversion was identical, within the experimental error, to glycerol carbonate and monoethyleneglycol yields, thus suggesting an almost perfect selectivity to the products (no or very minor presence of by-products).

Comment 4

“The zinc stearate catalyst is particular to the reaction between glycerol and ethylene carbonate producing glycerol carbonate and mono-ethylene glycol. If not, the author should compare the performance with other reported catalysts.”

Thank you for the comment. In fact, the zinc stearate is a mild basic catalyst also used in transesterification reactions. However, we find that, now, paragraph 3 in the Introduction section can serve as a fine comparison between zinc stearate and other basic catalysts for the tested reaction. Moreover, the reviewer can find a final paragraph in the Results & Discussion section comparing zinc stearate to other homogeneous catalysts in terms of TOF.

Comment 5

“Some small writing mistakes exist in the manuscript. The authors should carefully check and correct them.”

Thank you for the comment. We have gone through the text again, fixing all errors and typos we have found.

Reviewer 2 Report

The authors have synthesized the glycerin carbonate from ethylene carbonate using zinc stearate as catalyst under different operating conditions and applied and validated 3 kinetic models. This paper is well-structured and the topic is interesting as the author has collected sufficient literature regarding the presented topic which may helpful as a reference in this regard.

----L14-19…may not suit as Abstract and should be moved to the Introduction section at the end justifying the current study.

----“on the grounds of 23 results obtained at 80 °C and several catalyst concentrations and reagent molar ratios” The sentence in the abstract need revision to convey the intended meaning in a better way.

----The introduction section needs to be summarized especially couple of paragraphs with L58-79 and L87-104, thus combining both into one paragraph.

-----Authors should present briefly the materials/methodology of the study before the results and discussion.

----- L126-134 should be moved to Introduction/Methodology section, as appropriate.

----- L140-142…..same as above…..L168-171.

----- Is there any valid reason to select molar ratio with employing values of 1.5, 2, 2.5 and 3 ?

---- The Conclusions of the current findings should be presented for key take-home message.

Author Response

Second reviewer

“The authors have synthesized the glycerin carbonate from ethylene carbonate using zinc stearate as catalyst under different operating conditions and applied and validated 3 kinetic models. This paper is well-structured and the topic is interesting as the author has collected sufficient literature regarding the presented topic which may helpful as a reference in this regard.”

Thank you very much for your kind appreciation of our work.

Comment 1

“----L14-19…may not suit as Abstract and should be moved to the Introduction section at the end justifying the current study.”

We have rewritten in a more succinct manner these phrases in the Abstract section, while moving the main message to the end of the introduction, devoted usually to the aim of the work, as kindly suggested by the reviewer.

Comment 2

“---“on the grounds of 23 results obtained at 80 °C and several catalyst concentrations and reagent molar ratios” The sentence in the abstract need revision to convey the intended meaning in a better way.”

Thank you for your indication. We have rewritten the phrase in the abstract section. Now it reads as follows:

“Subsequently, an appropriate kinetic model was fitted to all data obtained at 80 °C and several catalyst concentrations and reagent molar ratios.”

Comment 3

“----The introduction section needs to be summarized especially couple of paragraphs with L58-79 and L87-104, thus combining both into one paragraph.”

We appreciate very much the comment of the reviewer. Following it, we have shortened both paragraphs and joined them in only one paragraph. Now it reads as follows:

“To carry out this transcarbonatation reaction, several basic catalysts have been tested, among which the following can be mentioned: KOCH3 [5] K2CO3 [8], CaO [9], and Al–Mg and Al–Ca hydrotalcites [10]. Potassium methoxide is a strong basic catalyst shows a very high activity in biphasic media at 40-50 °C at 50-150 ppm catalyst in solventless conditions, as it concentrates in the glycerol phase [5]. Potassium carbonate is also a homogeneous catalyst with lower basicity; reactions proceed at mild temperatures in solventless conditions and higher catalyst concentrations (till 1.25% w/w of glycerol) [8]. Classical heterogeneous catalysts are calcium oxide and Al-Mg hydrotalcites. CaO performed well under the following reaction conditions: GL: 9.2 g; DMC/Gly molar ratio: 3; reaction temperature: 80 °C; Cat/GL molar ratio: 0.15 (in CaO weight); at a reaction time of 5 hours, glycerol conversion was more than 95% while the carbonate yield was 90.57%. However, it shows a notable deactivation surely due to water action. As for the hydrotalcites, using a 2 to 1 molar ratio of EC to glycerol, 7 %(w/w) of catalyst respect to the total weight of reactants, and a temperature of 50 °C, yield to glycerol carbonate was 82% with only minor amounts of glycidol at 2 h [9]. De-activation is evident after only three cycles. Considering metallic soaps, zinc carboxylate was studied as an active catalyst in the triglyceride transesterification reaction to produce biodiesel [11]. With this catalyst, triglyceride conversion reached values high-er than 80% at 60 min, 5% catalyst concentration and 10-30 methanol/oil molar ratio; FAMEs yield was near 60% at those conditions, but only at the highest methanol/oil molar ratio. This catalyst was also used as catalyst for the esterification of fatty acids (a commercial mix of oleic acids) with ethanol, as was zinc laurate, reaching acid conver-sions up to 96% in 90 minutes [12]. In that case, the catalyst could be recovered after the reaction as a powder by simple filtration and used again with no activity loss due to its layered structure. Zinc stearate is also useful when employed in a free fatty-acid rich olive pomace oil, reaching very high conversions if glycerol and water is progressively removed [13,14] Alkali metal carboxylates have also shown to be highly active as catalysts for the ring-opening polymerization of cyclic esters [15,16]. To our best knowledge, however, there are no reports on the activity of metal fatty acid salts acting as catalyst to produce glycerol carbonate or, in fact, in any transcarbonatation reaction.”

Comment 4

“-----Authors should present briefly the materials/methodology of the study before the results and discussion.”

Thank you for the indication. Due to the structure of the journal template, the Materials and Methods section (section 3 in this case) is placed after the Results and Discussion section (section 2 in this case).

Comment 5

“----- L126-134 should be moved to Introduction/Methodology section, as appropriate.”

We really appreciate the comment. However, on one hand, we cannot see the line numbers in the word file provided to us, and, on the other, we have provided some information at the beginning of each subsection related to the experiments performed or giving a general idea of the importance of what is done in such subsection. We feel that this is needed to guide the reader, especially in the case con the subsection devoted to mass transfer.

Comment 6

“----- L140-142…..same as above…..L168-171.”

It is the same case, I fear, as in the previous comment. We cannot identify the lines, due to the format of the document we have, but, in any case, the information is just intended for the benefit of the readers.

Comment 7

“----- Is there any valid reason to select molar ratio with employing values of 1.5, 2, 2.5 and 3?”

Usually, an excess of the carbonate reagent is needed to shift the equilibrium towards the targeted carbonate. However, in our experience, an excess can promote, depending on the selectivity of the catalyst, a polymerization or oligomerization reaction. In any case, the reviewer can see in the Introduction section how several authors choose carbonate/glycerol molar ratios around 2-3.

Comment 8

 “---- The Conclusions of the current findings should be presented for key take-home message.”

Thank you for the observation; we have not included it due to the low number of pages of this article; though, the authors agree with the reviewer that this type of section is almost always welcome. We have included the following remarks or messages to take home in a dedicated Conclusion section:

“3. Conclusions

In the transcarbonatation reaction between glycerol and ethylene carbonate driven by zinc stearate, we can observed the typical hyperbolic trend of the initial reaction rate at increasing amounts of the catalyst, indicating the interfacial nature of the chemical transformation. An excess of the carbonate reagent involves an increase in the final yield but, at reagent molar ratios higher than 2, this excess leads to a decrease of the initial reaction rate, indicating an excessive dilution of the glycerol. The trend of the initial reaction rate with temperature is exponential, as expected, while the two-stages kinetic modelling suggests that the reaction is bimolecular and elemental and that the process is interfacial in essence, with the catalyst dispersed in a narrow space between a polar (glycerol) and a nonpolar (ethylene carbonate) phases. Furthermore, a molar ratio of 2 or higher seems to promote a good dispersion of glycerol in ethylene carbonate, increasing final yields.”

We are very grateful to the reviewer for his/her inspiring comments that have notably help us to improve the manuscript.
